# Full thermoelectric characterization of a single molecule

Andrea Gemma[1], Fatemeh Tabatabaei[2], Ute Drechsler[1], Anel Zulji[1], Hervé Dekkiche[3], Nico Mosso[1], Thomas Niehaus[2], Martin R. Bryce [3], Samy Merabia [2] & Bernd Gotsmann [1] ✉

Molecules are predicted to be chemically tunable towards high thermoelectric efficiencies and they could outperform existing materials in the field of energy conversion. However, their capabilities at the more technologically relevant temperature of 300 K are yet to be demonstrated. A possible reason could be the lack of a comprehensive technique able to measure the thermal and (thermo)electrical properties, including the role of phonon conduction. Here, by combining the break junction technique with a suspended heat-flux sensor, we measured the total thermal and electrical conductance of a single molecule, at room temperature, together with its Seebeck coefficient. We used this method to extract the figure of merit $zT$ of a tailor-made oligo(phenyleneethynylene)-9,10-anthracenyl molecule with dihydrobenzo[*b*]thiophene anchoring groups (DHBT-OPE3-An), bridged between gold electrodes. The result is in excellent agreement with predictions from density functional theory and molecular dynamics. This work represents the first measurement, within the same setup, of experimental $zT$ of a single molecule at room temperature and opens new opportunities for the screening of several possible molecules in the light of future thermoelectric applications. The protocol is verified using SAc-OPE3, for which individual measurements for its transport properties exist in the literature.

The transport of heat and charge through molecular junctions exhibits rich transport physics[1,2]. A better understanding of transport phenomena at this scale can lead to important progress in different fields, from power management in current computer architectures, to energy conversion. Careful experiments have enabled the study of molecular junctions at the level of single molecules, opening the way to the exploration of the limits of charge and phonon transport through single channels at the nanoscale[3–5].

Among the possible applications of molecular junctions, thermoelectric energy conversion has probably received the largest attention in recent research, as molecules hold the promise for a dramatic increase to the efficiency of heat-to-charge conversion. This is based on the possibility of atomically tuning their chemical structure, introducing novel features, like phonon suppression and quantum interference, that are not possible in standard thermoelectric materials[6–8].

For this reason, many new and different molecules are currently fabricated, and their thermoelectric properties are extensively studied, both theoretically and experimentally. However, most studies in the literature focus on simulating, fabricating, and testing molecules in order to obtain the best possible results in terms of Seebeck effect, which is directly responsible for converting heat into electrical energy. However, to achieve an effective conversion, the temperature drop sustained between the ends of the molecule should be large enough to induce an appreciable charge current, requiring simultaneously a low level of thermal conductance and a high level of electrical

[1]IBM Research Europe – Zurich, 8803 Rueschlikon, Switzerland. [2]Université Claude Bernard Lyon 1, CNRS, Institut Lumière Matière, F-69622 Villeurbanne, France. [3]Department of Chemistry, Durham University, Durham DH1 3LE, UK. ✉e-mail: bgo@zurich.ibm.com

conductance[9]. Thus, a more comprehensive way to evaluate the efficiency of a molecule to convert heat into electricity is to consider the so-called material thermoelectric figure of merit $zT$. When the temperature difference $\Delta T$ occurring across the molecular junction is not too large, $zT$ can be expressed as:

$$zT = \frac{S^2 G}{\kappa} \cdot T \qquad (1)$$

where $G$ is the electrical conductance, $T$ is the average temperature across the two leads, and $\kappa$ is the total thermal conductance which includes both contributions coming from electrons and phonons ($\kappa_{el}$ and $\kappa_{ph}$, respectively).

Experimental values of $zT$ are hardly available in the literature, perhaps due to the difficulty in experimentally measuring the total thermal conductance $\kappa$. At the single-molecule level, the only experimental measurement of molecular $zT$ required 2 K absolute temperature and the fitting of a theoretical model to extract the thermal conductance of the molecule[10]. At the technologically more relevant temperature range around 300 K, only alkanedithiol and an oligo(phenyleneethynylene) (OPE3) have been characterized in terms of their thermal conductance[4,5]. A demonstration of an experimental setup and protocol to fully characterize the thermoelectric figure of merit of single molecules has been lacking, in particular for operation at ambient temperatures and with the capability to account for the phononic contribution to the thermal conductance. It is important to combine the methods for measuring $G$, $S$, and $\kappa$ for a single setup, because of the statistical character of the measurements of molecular junctions using break-junction techniques. In these techniques the repeated opening and closing of a junction is used to determine a likely arrangement of a molecule between two metal electrodes by means of statistical averaging of measurement data. To combine such measurements from different setups is challenging and requires a high level of reproducibility.

In this work, we report on an exhaustive method to fully characterize experimentally the thermoelectric figure of merit of an oligo(phenyleneethynylene)-9,10-anthracenyl single molecule junction with dihydrobenzo[$b$]thiophene anchoring groups (DHBT-OPE3-An), by measuring its electronic, thermal, and thermoelectric properties at room temperature. Given the very tedious and slow thermal transport measurements, this particular molecule was chosen as to address some of the most important current questions in the field. Namely, the interesting aspect of this molecule is the increase of vibrational states by locally adding atoms to only one (the central) part of the molecule with respect to the reference OPE3 molecule. This is an important step towards the inclusion of moieties to enhance or reduce the thermal conductance of a molecule through localization and quantum interference effects[1]. Furthermore, the attachment of side-groups in the central part of a molecule is a starting point of cross-linking strategy of molecular layers[11]. In addition, we study, in the thermal context, the enhanced binding yield reported for DHBT end groups, which may reduce the systematic risk of statistical approaches. These data then motivate further studies with series of molecules systematically varying only one aspect, which is commonly done for the simpler electrical conductance and thermoelectric measurements.

To verify the reliability of thermal transport studies, a comparison between the experimental and theoretically expected values for the thermal conductance is given, as well as a benchmark with respect to similar OPE3-based molecules. Our results represent, to our knowledge, the first ever experimental measurement of the complete $zT$ of a single molecule at room temperature and within the same setup, including the phononic contribution to the total thermal conductance. In addition, the protocol is verified using SAc-OPE3, for which individual measurements for its transport properties exist in the literature.

## Results

### Sample preparation

Molecules are deposited on the gold platform by dip coating, with concentrations ranging from 0.1 to 1 mM, for 30 s to 2 h. After the deposition, samples are rinsed several times in clean solvent to wash away physically adsorbed molecules. The synthesis and purification of the molecule under study (DHBT-OPE3-An) are described by Dekkiche et al.[11].

The thermal transport measurements require a sample cleaning procedure compatible with the under-etched sensor structure described below. The post-fabrication cleaning procedure for the MEMS is a combination of oxygen plasma and ion milling to remove contaminants and retrieve a fresh gold surface.

### Charge, heat, and thermoelectric transport measurements

The experimental setup and procedure are shown schematically in Fig. 1. All the measurements are performed in high vacuum ($\sim 10^{-7}$ mbar), at room temperature, and within a custom-built scanning tunneling microscope (STM), located in one of the IBM Noise Free Labs[12].

We follow the measuring protocol to extract the electrical and thermal conductance of a single molecule described by Mosso et al.[4]. Compared to other examples in the literature[13,14] this protocol adds the possibility to measure the thermal conductance together with the electrical conductance, and it has been validated against two model systems, namely, dithiol-oligo(phenylene ethynylene) and octane dithiol junctions with gold electrodes. The results are in good agreement both with theory and other independent studies[5]. Briefly, to measure simultaneously the electrical and thermal transport properties of a single-molecule junction, a suspended micro-electro-mechanical system (MEMS) has been devised[15], featuring a low thermal conductance $\kappa_{MEMS}$ ($3.5$–$4.5 \times 10^{-8}$ W/K), Fig. 1a. The device comprises a Pt heater/thermometer, and a central membrane with a gold platform, which can be used to form electrical contacts with the tip of a STM. The membrane is suspended by means of four silicon nitride beams.

### Electrical conductance measurements

The experiment is based on performing STM-Break Junction (STM-BJ) measurements, where an electrochemically etched gold tip is brought into and out of contact with the gold platform on top of the suspended MEMS. The electrical conductance ($G$) between the tip and the platform is continuously monitored during contact formation (closing trace) and contact breaking (opening trace). When looking at the evolution of $G$ versus electrodes separation, it is sometimes possible to observe a plateau below the value of the conductance quantum ($G_0$). This usually indicates the presence of a molecule stretching inside the junction[16–18]. At the end of such a plateau the molecular contact is then eventually broken. Figure 1b shows an example of a single opening trace at a retraction speed of 1 nm/s, where two different regimes (in contact/out of contact) can be distinguished.

For the measurement discussion of the electrical conductance of the molecular junction, we consider the most recurrent value of the molecular plateau (i.e., the maximum of the molecular peak in the 1D histogram in Fig. 2a).

### Thermal conductance measurements

For measuring the thermal conductance of the molecular junction, the temperature of the suspended membrane is first increased to $T = T_H$ (from 10 to 60 K above room temperature), by applying a constant voltage (usually 50 or 90 mV) to the Pt-heater (typical resistance ~20 kΩ), corresponding to few μW of dissipated power. The temperature $T$ as function of electrodes separation is then continuously monitored, by reading the four-probe electrical resistance of the Pt-heater. The slow retraction speeds on the order of nm/s enable the measurement of thermal measurements described below, which have

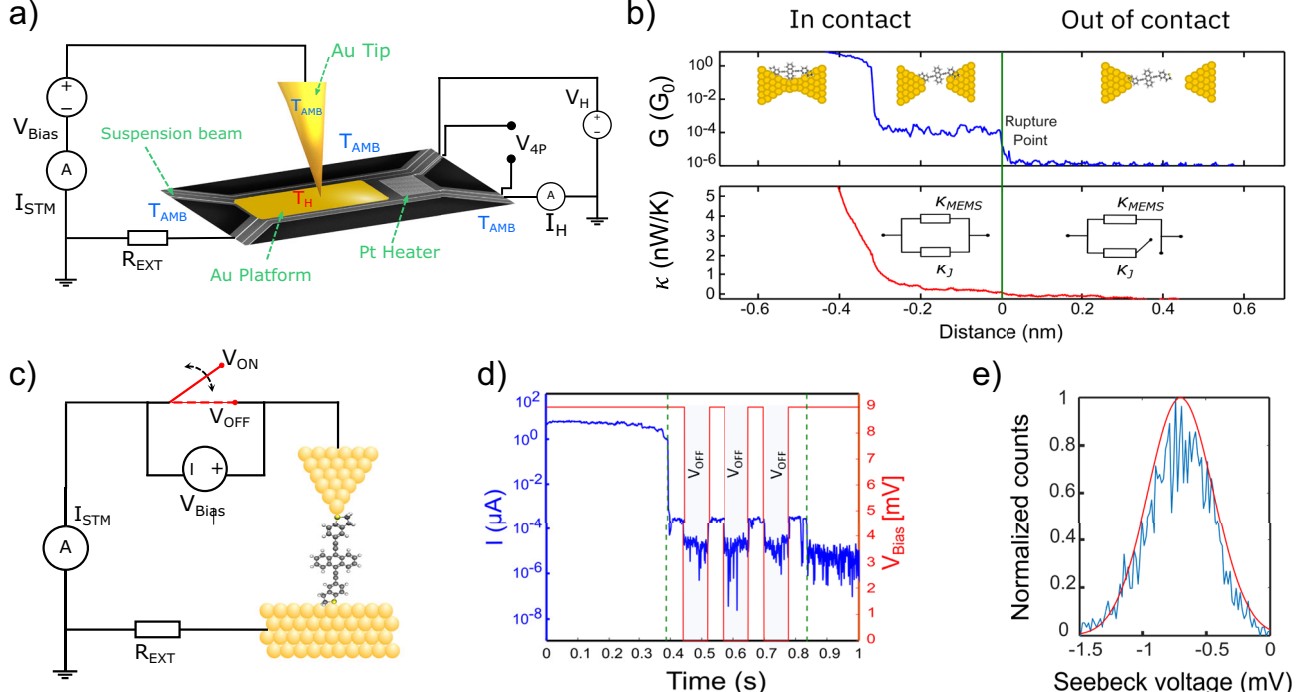

**Fig. 1 | Experimental setup and measurement protocol. a** Schematic representation of the MEMS structure used in the differential measurement of the junction thermal conductance $\kappa_J$. **b** The electrical conductance, $G$, is used as reference (green vertical line) to find the rupture point of the molecular plateau. The thermal conductance is then obtained as the difference between the values before and after the rupture point. **c** Schematic representation of the technique used for the thermoelectric characterization of a single molecule junction. The red switch represents the biasing status of the tip. When no electrical bias is applied, the measured current consists of only the thermoelectric contribution; **d** typical example of an opening trace: $I(t)$ and $V_{Bias}(t)$ for a single-molecule junction. The green-dashed lines represent the breaking points for the single atom and single-molecule junctions, respectively; **e** typical Seebeck voltage measured across the junction when $V_{Bias} = 0$, here at $\Delta T = 40$ K. The width of the distribution is given by the different configurations the molecule can assume inside the junction.

a limited measurement bandwidth given by the thermal time constant of the MEMS sensor in the range of tens of ms.

At the beginning of an opening trace, when the tip is in contact with the membrane, the total thermal conductance of the system is given by $\kappa = \kappa_{MEMS} + \kappa_J$, where $\kappa_{MEMS}$ represents the parasitic contribution to the thermal conductance through the suspending beams, while $\kappa_J$ is the conductance of the junction under study. After breaking the molecular junction, the only contribution to the total thermal conductance is represented by the parasitic $\kappa_{MEMS}$, Fig. 1b. The difference between the in-contact value and the out-of-contact value gives the thermal conductance $\kappa_J$ of the junction. We note, the junction thermal conductance $\kappa_J$ is so small that it dominates the entire thermal path even though the already low values of $\kappa_{MEMS}$. Therefore, thermal four-probe techniques as described for example in ref. 19 can be avoided. On a molecular scale no convincing approach for a four-probe measurement (neither electrically nor thermally) has been proposed. One therefore talks about the junction conductance rather than the conductance of a molecule.

During the course of the experiments the junction goes through the many possible binding configurations a molecule can assume between the two electrodes. Similar to what has been established for electrical conductance measurements, a statistical approach is used to determine the most probable configuration. We typically collect 3000–5000 traces per data set, out of which some hundreds showing a clear molecular signature. (It is important to limit the number of such "successful" events by means of controlling the molecule concentration, to be sure that events with multiple molecules contacted at the same time can be minimized.) With those traces, 2D histograms for the electrical conductance (Fig. 2a) and thermal conductance (Fig. 2b) are built by aligning all of the traces at the breaking point of the molecular junctions[4].

To extract the thermal conductance of such junctions ($\kappa_J$), we take the difference between the two linear fits of the median in the thermal histogram before and after the molecular breaking point, see Fig. 2b. The small variations of the slopes extracted from the fitting of different data sets are not systematic and are encompassed within a certain variation. In the small fitting range of a few ångströms, higher order fits did not yield better results.

**Thermoelectric transport measurements**

The experimental procedure for the measurement of the Seebeck coefficient is represented schematically in Fig. 1c–e), and follows the same protocol as presented by Dekkiche et al.[11] This protocol has been applied also to OPE3 with SAc end groups, a molecule that has been studied using the same setup in thermal conductance measurements (see supplementary information). In a typical experiment, the temperature bias between the two electrodes is first adjusted to have an accuracy and stability better than 0.01 °C. The overall temperature difference, $\Delta T$, between the two electrodes ranges from 0 up to 60 K. All measurements are performed in high vacuum and at room temperature. Once thermal equilibrium has been reached, the tip is moved closer to the counter electrode until a good electrical contact has been established ($G > 5\,G_0$). Then the two electrodes are slowly moved apart (opening trace), usually at a speed of 3 nm/s. A typical opening trace for the Seebeck measurement is shown in Fig. 1d.

Since for the DHBT-OPE3-An molecule, a conductance plateau of around $10^{-4}\,G_0$ is predicted, a molecular region is defined in the interval of conductance between $10^{-3}\,G_0$ and $10^{-5}\,G_0$. In this region, the speed of the piezoelectric scanner is lowered to 1 nm/s.

When a molecular plateau is found within the boundaries of such molecular region, the conductance $G_{before}$ of the first 50 points of the plateau is measured. If the value stays between the boundaries

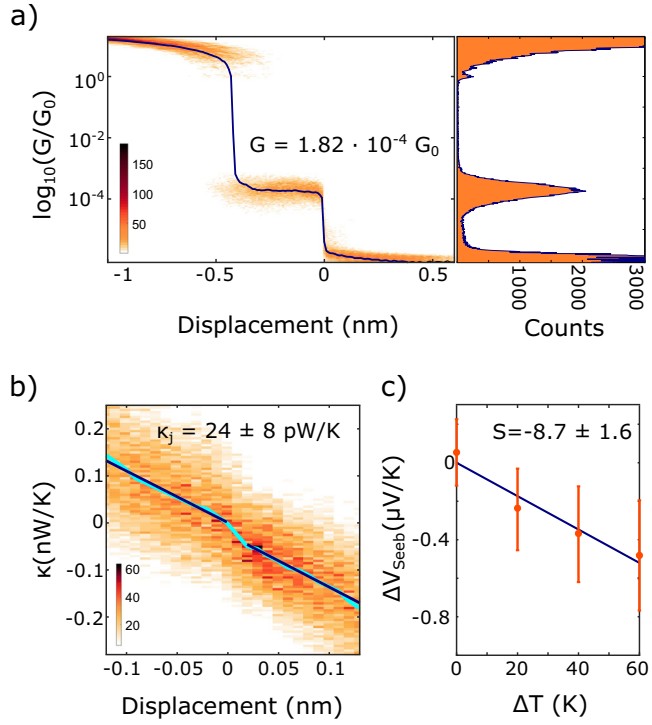

a)

b)

$\kappa_j = 24 \pm 8$ pW/K

c)

S=$-8.7 \pm 1.6$

**Fig. 2 | Experimental thermoelectric characterization for a single DHBT-OPE3-An junction obtained at room temperature. a** Left side: 2D histogram of the electrical conductance $G$ vs. displacement (opening trace), obtained from 297 traces. The dark blue line is the median of the histogram (logarithmic binning). Right side: 1D histogram of the electrical conductance for the same molecule. The maximum of the molecular peak is located at $1.82 \times 10^{-4}$ $G_0$. **b** 2D histogram of the thermal conductance $\kappa$ vs. displacement (opening trace), measured with $\Delta T = 20$ K. The pale blue line is the median of the 2D histogram. The dark blue lines are the two fits before and after the molecular rupture point. **c** Seebeck voltage vs. applied temperature differences $\Delta T$. The slope of the blue line is the Seebeck coefficient (also called thermopower), the error bars indicate the width of the Seebeck voltage distribution for the given $\Delta T$. **a** and **c** are adapted from ref. 11, with permission from the Royal Society of Chemistry.

$(10^{-3}$ $G_0$–$10^{-5}$ $G_0)$, it is assumed a molecule is inside the junction. Afterward, the electrical bias $V_{Bias}$ is turned off for a window of 80 datapoints (the switch in Fig. 1c is closed), and the thermoelectrical current $I_{Seeb}$ is measured. At this point, the electrical bias is restored and the conductance $G_{after}$ of the following 50 datapoints is compared to the value before the switching. If both values of conductance, $G_{after}$ and $G_{before}$, are found to be within the thresholds of the molecular region and $G_{after} = G_{before} \pm 20\%$, the switching is repeated until the previous conditions on the conductance are no longer fulfilled.

A thermovoltage, $\Delta V_{Seeb}$, is then calculated from the measured thermocurrent using the following equation:

$$\Delta V_{Seeb} = \frac{I_{V_{Bias}=0}}{G_{avg}} = S\Delta T \qquad (2)$$

where $G_{avg}$ is the average between $G_{before}$ and $G_{after}$. We note that our scheme requires a switching of applied bias without inducing a rupture of the junction. Other protocols to determine $S$ in molecular junctions measures directly the Seebeck voltage at zero current[2]. This has the disadvantage of not monitoring the state of the junction through a measurement of $G$, or having to switch the current path, which we found more difficult to realize experimentally.

The Seebeck voltage is finally plotted for different values of $\Delta T$, as shown in Fig. 2c. The slope of the line fitting the datapoints, is an experimental measurement of the Seebeck coefficient of the single-molecule junction. For each datapoint, at least 500 measurements are considered. The error bar of each datapoint in Fig. 2c indicates the width of the relative $\Delta V_{Seeb}$ distributions (Fig. 1e). We stress that the measurement noise is smaller than the width of the distribution, which is related to probing different configurations that the molecule can assume inside the junction.

## Experimental results

As can be seen from Eq. 1, the Seebeck coefficient ($S$), the electrical conductance ($G$), and the thermal conductance ($\kappa$) of a single molecule need to be measured to get experimentally the full $zT$. For the molecule under study the electrical conductance and Seebeck coefficient have already been reported in Dekkiche et al.[11], while the thermal conductance $\kappa$ is reported for the first time here in Fig. 2b.

Figure 2b shows in pale blue the median of the underlying 2D thermal histogram, displaying a noticeable step in the thermal background at the breaking point of the molecular junction ($x = 0$ nm). The difference at 0 between the linear fits yields a molecular thermal conductance value. Averaging over two independent measurements (Fig. 3) we extract $\kappa = (23.5 \pm 6.9)$ pW/K. The agreement between the two independent measurements confirms the repeatability of the experiment.

By comparing the experimental value with the theoretical predictions for the phononic thermal conductance through the junction, it can be seen that most of the heat transported across the molecule is carried by phonons, while the electronic contribution only plays a negligible role (Fig. 3).

## Theoretical calculations

To construct a theoretical picture of thermal transport across the molecular junction, we employ two simulation methods: molecular dynamics (MD) and a combination of techniques based on density functional theory. The former method allows us to estimate the phonon thermal conductance while the latter method serves to estimate the electronic thermal conductivity. For electronic transport calculations, the molecular device has been constructed in several steps as detailed in Dekkiche et al.[11]. First, using the NWCHEM package[20], we performed all-electron calculation to optimize DHBT-OPE3-An in the gas phase. Then, using the Atomic Simulation Environment (ASE), the optimized DHBT-OPE3-An was placed between two Au (111) surfaces containing six layers with 144 gold atoms per layer. The electronic thermal conductivity is estimated using a combination of non-equilibrium Green's functions (NEGF) and Density Functional Tight Binding (DFTB) calculations as described in refs. 21,22. In particular, the value of the electronic thermal conductivity at 300 K was calculated using the transmission function of the device, as described in ref. 23.

In parallel, MD simulations have been performed using the LAMMPS package. The interactions inside the molecule are described by the OPLS force field[24], while gold is simulated with the Heinz potential[25]. The gold-sulfur interaction, instead, is described by the Morse potential[21]. The molecular junctions are first thermalized for 1 ns in the NPT ensemble and then non-equilibrium MD simulations are employed to extract the thermal conductance. In these latter simulations, the two gold reservoirs were respectively heated up and cooled down by 50 K and the thermal conductance was estimated based on the accumulated energy of the hot and cold gold reservoirs. The results presented here were obtained after averaging over 30 independent simulations each representing a total duration of 1 ns.

From the electronic transport calculations, we first deduce the value of the electronic thermal conductivity $\kappa_{el} = 0.456$ pW/K. This value is more than 50 times smaller than the value of the phonon thermal conductance obtained in MD: $\kappa_{ph} = 23.7 \pm 1.3$ pW/K. The predicted thermal transport across the molecule junction of $\kappa_{TOT} = 24.2$ pW/K is therefore dominated by phonon degrees of freedom, Fig. 3.

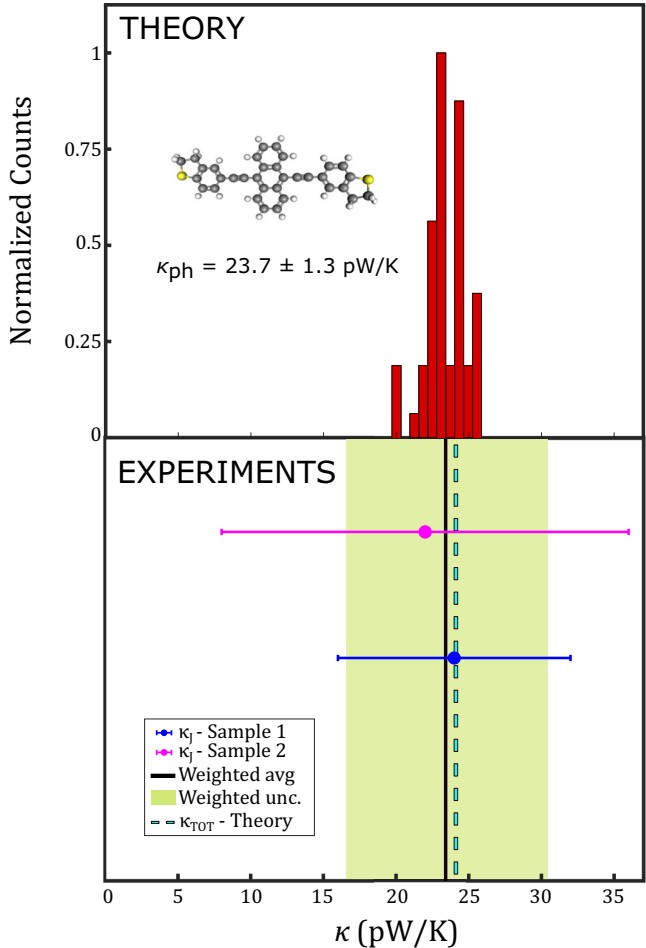

**Fig. 3 | Histogram of the theoretical phononic contribution $\kappa_{ph}$ to the total thermal conductance of the junction (top) vs. experimental total thermal conductance $\kappa_J$ (bottom).** In the top panel, the results obtained from MD simulations for the phononic thermal conductance of the DHBT-OPE3-An molecule are represented in the form of a histogram. The structure of the DHBT-OPE3-An molecule is also shown (carbon atoms in gray, hydrogen atoms in white, sulfur atoms in yellow). In the bottom panel, the black solid line is the weighted average (avg) between the two experiments, represented here by the two circles (sample 1, sample 2), each with their own uncertainty. The green shaded area, instead, represents the uncertainty (unc.) on the weighted average. For comparison with the experimental values, the cyan dashed line represents the total predicted thermal conductance $\kappa_{TOT}$ obtained as a sum of the theoretical phonon conductance $\kappa_{ph}$ (i.e., the average of the theoretical histogram on top) plus the theoretical electronic contribution $\kappa_{el}$ predicted by means of density functional theory.

We conclude that the thermal conductance computed in MD is in excellent agreement with the value determined experimentally. For comparison, we also computed the phonon thermal conductance of a DHBT-OPE3-Ph molecule, which is structurally similar to the DHBT-OPE3-An but has only a simple phenylene group in the central part of the molecule instead of the anthracene. The value extracted there was $\kappa_{ph} = 21.99 \pm 3.80$ pW/K.

## Discussion

The results represented in Fig. 2 show the ability of our setup to perform a complete thermoelectric characterization of a single molecule junction and to experimentally determine its full room temperature $zT$. This is a long-awaited result in the field of molecular thermoelectricity

since it allows for the experimental screening of different molecules in vision of viable applications.

For a single DHBT-OPE3-An, we measured the electrical conductance $G$ (Fig. 2a), the thermal conductance $\kappa$ (Fig. 2b), and the Seebeck coefficient $S$ (Fig. 2c). By combining the three measured quantities into Eq. 1, we get:

$$zT \simeq 1.3 \times 10^{-5}$$

The protocol and setup described here has been verified carefully in its individual measurements. In particular, the thermal and electrical conductance measurement technique was previously independently verified by two groups 4,5, and the Seebeck coefficient measurement was verified using SAc-OPE3 in comparison to literature (see supplementary information). The SAc-OPE3 molecule yields $zT \simeq 2 \times 10^{-5}$. These $zT$ values still fall short for what is actually needed to realize a marketable molecular thermoelectric converter[9]. However, the molecule of this study was specifically chosen to address some of the current important scientific questions, where experimental confirmation was needed.

First, with respect to the more commonly used thiol anchor groups, this variant of OPE3 links to the gold electrodes with DHBT anchor groups. The previously reported high junction yield[11] was again observed here, supporting this anecdotal evidence. Further, this result confirms that the choice among these anchor groups has a relatively small influence on the phonon conductance. This is relevant, because theoretical predictions of thermal conductance of OPE3 molecules with different anchor groups cover a significant range between 19 and 34 pW/K[4,26,27]. In this context, it appears that, within the accuracy of the experimental and theoretical results presented here, the DHBT-OPE3-An possesses a thermal conductance that is slightly lower than average despite the improved cleanliness of the binding.

Secondly, within our experimental uncertainty, this work also confirms that phonon engineering is possible in single molecules. Indeed, it is interesting to notice that despite an increase of 30% in the number of atoms and degrees of freedom, the thermal conductance of the DHBT-OPE3-An only increases by about 10% compared to the predicted 22 pW/K for the DHBT-OPE3-Ph analog, hinting at the influence of side groups. The reason could be the localized nature of vibrational modes within the molecule, which do not equally contribute to the coherent transport along the junction. In the context of potential technological use, the ability to readily attach side groups onto the central ring of OPE3 could be a way to independently tune thermal and thermo-electrical transport[1,28], as well as to provide options for cross-linking of molecular junctions[11].

## Data availability

Source data are provided with this paper.

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

## Acknowledgements

The authors gratefully acknowledge support from the Cleanroom Operations Team of the Binnig and Rohrer Nanotechnology Center (BRNC) for their help and technical support. Special thanks go to A. Molinari for the useful discussions and to E. Vidal-Revel for the help in the lab. This work has received funding from the European Community through the Horizon 2020 Research and Innovation Programs under Grant Agreement Nos. 767187 (QuIET) and 766853 (EFINED), the FP7 ITN "MOLESCO" No. 606728, and through the Swiss National Science Foundation under Grant No. 200660. Special thanks go to K. Weiland and M. Mayor for the purification step of the SAc-OPE3 molecules.

## Author contributions

A.G., N.M., and B.G. designed the thermoelectric experiment and the setup, with help from U.D. and A.Z.; A.G. performed the experimental work and data analysis with inputs from B.G. and N.M.; F.T., T.N., and S.M. carried out the theoretical calculations; H.D. and M.R.B. designed the molecule which was synthesized and purified by H.D.; A.G. and B.G. wrote the manuscript with input from all co-authors.

## Competing interests

The authors declare no competing interests.
