## [Peer Review File · Nature Communications]

REVIEWER COMMENTS

Reviewer #1 (Remarks to the Author):

The manuscript proposed a method to measure the thermal and electrical conductance of a single molecule combining the break junction technique with a suspended heat-flux sensor. And the zT of an oligo(phenyleneethynylene)-9,10-anthracenyl single molecule was measured at room temperature. I think this manuscript can be published after revisions.

1. The developed measuring protocol to evaluate the electrical and thermal conductance of single molecules should be compared with that reported in literatures.
2. More molecules except the oligo(phenyleneethynylene)-9,10-anthracenyl derivative should be tested to verify the designed protocol.
3. The format of the manuscript is mess, such as some words were aligned at both ends and some at left end. Background literature is somewhat missing, for example Nat. Commun. 2016, 7, 11294.
4. The words in Figure 3 were not clear.

Reviewer #2 (Remarks to the Author):

The author has measured the thermal and electrical of single molecule, as well as the Seebeck coefficient at room temperature. Based on the method, the ZT value is calculated. However, the manuscript contains several serious issues that need to be addressed, and major points are listed as follows:

- 1) There are typos should be carefully checked and revised, for example, the subtitle should be "introduction"; the Line 173, what does the 'F' represent for, and so on.
- 2) why the author choses the oligo(phenyleneethynylene)-9,10-anthracenyl single molecule junction with dihydrobenzo[b]thiophene anchoring groups to do the calculation and get get experimentally the full zT . Can this method be applied to another general molecules?
- 3) The author should clarify the novelty of this manuscript and what developments of science is this manuscript contributed to.
- 4) The author claimed the experimental result is in excellent agreement with density functional theory and molecular dynamics predictions. More experimental details should be given in the manuscript, otherwise, it is insufficient to judge whether the method is reasonable or not.

Reviewer #3 (Remarks to the Author):

The authors demonstrated an alternative technique to simultaneously measure the thermal and electrical properties at room temperature with a small molecule for thermoelectric application. The experiment is well designed, and the results which are argued by the authors are impressive.

However, a suspended heat-flux sensor seems to be a general method; Phys. Rev. B 2011, 83, 113305. A break junction technique has been explained in the STM research field.

Moreover, the material used in this research is already reported in Nanoscale, 2020,12, 18908-18917.

Even though the performance has an impact on the thermoelectric characterization of a single molecule, the concept is not new. Therefore, I can't entirely agree this research meets the criteria of this highly reputational journal. The authors should clarify the following issues to strengthen this manuscript.

- 1) The authors need to clarify the concept of the simultaneous measurement in thermoelectric parameters with a detailed explanation compared to Phys. Rev. B 2011, 83, 113305.
- 2) To confirm the system represented in this research, they show the data with another single thermoelectric molecule.
- 3) In terms of the lattice contribution of the thermal conductivity, they need to represent the experimental data compared with MD simulation.

Reviewer: 1

Comment 1

The developed measuring protocol to evaluate the electrical and thermal conductance of single molecules should be compared with that reported in literatures

Reply 1

We thank the reviewer for the comment, and we agree that a better comparison of the method with the previous literature is needed.

We added to the manuscript with respect to the thermal measurements:

“Compared to other examples in the literature^{12,13} this protocol adds the possibility to measure the thermal conductance together with the electrical conductance, and it has been validated against two model systems, namely, dithiol-oligo(phenylene ethynylene) and octane dithiol junctions with gold electrodes. The results are in good agreement both with theory and other independent studies⁵.”

We also added a comparison to other protocols for the measurements of the Seebeck voltage:

*“We note that our scheme requires a switching of applied bias without inducing a rupture of the junction. Other protocols to determine S in molecular junctions measures directly the Seebeck voltage at zero current [Cui et al., *The Journal of Chemical Physics* 146 (9): 092201, (2017)]. This has the disadvantage of not monitoring the state of the junction through a measurement of G , or having to switch the current path, which we found more difficult to realize experimentally.”*

Comment 2

More molecules except the oligo(phenyleneethynylene)-9,10-anthracenyl derivative should be tested to verify the designed protocol.

Reply 2

We thank again the reviewer for the comment, which has been brought up in a similar manner also by another reviewer. Indeed, we fully agree that in the area of molecular electronics it is now common to use a series of molecules with a systematic variation of a certain property to draw conclusions. This is common for both electrical and thermoelectric transport measurements. We fully agree that would be very advantageous to further verify the method and to gain more knowledge on such an interesting and relevant field as molecular thermoelectricity. However, due to the extremely small magnitude of the thermal signal originating from a single molecule junction, experiments are tedious and time consuming, with ultimate low yield. Since the advent of single molecule experiments, only six other molecules have been studied in terms of their thermal conductance at the single molecule level, with the one in this study being the seventh and first one after the pioneering results published in 2019 [cf. Cui et al. *Nature* **572**, 628–633 (2019) and Mosso et al. *Nano Lett.* **19**, 7614–7622 (2019)]. We explain this in the introduction.

We agree that the lack of access to more measurements requires a better explanation as to why this particular molecule was chosen. We added to the manuscript:

“Given the very tedious and slow thermal transport measurements, this particular molecule was chosen as to address some of the most important current questions in the field. Namely, the interesting aspect of this particular molecule is the increase of vibrational states by locally adding

atoms to only one (the central) part of the molecule with respect to the reference OPE3 molecule. This is an important step towards the inclusion of moieties to enhance or reduce the thermal conductance of a molecule through localization and quantum interference effects [Gotsmann et al. *Applied Physics Letters* 120 (16): 160503 (2022)]. Furthermore, the attachment of side-groups in the central part of a molecule is a starting point of cross-linking strategy of molecular layers [Dekkiche et al., *Nanoscale* 12 (36): 18908–17, (2020)]. In addition, we study, in the thermal context, the enhanced binding yield reported for DHBT end groups, which may reduce the systematic risk of statistical approaches. These data then motivate further studies with series of molecules systematically varying only one aspect, which is commonly done for the simpler electrical conductance and thermoelectric measurements.”

Given these limitations, we are still confident about the method, because individual parts of the protocol were confirmed or discussed in comparison to protocols described in the literature (independent measurement of thermal conductance of the standard OPE3 molecule by two groups as cited in the introduction, confirmation of electrical conductance measurements of benchmark molecules as cited in the introduction, and comparing the Seebeck protocol to others in literature as discussed in the revised manuscript).

Comment 3

The format of the manuscript is messy, such as some words were aligned at both ends and some at left end. Background literature is somewhat missing, for example Nat. Comm. 2016,7,11294

Reply 3

We thank the reviewer for reading our manuscript despite the formatting errors, which were fixed in the revision.

We also added more background literature to help readers. In particular, the following references have been added:

1. Cui, L., Miao, R., Jiang, C., Meyhofer, E. & Reddy, P. Perspective: Thermal and thermoelectric transport in molecular junctions. *J. Chem. Phys.* **146**, 092201 (2017).
2. Xu, B. & Tao, N. J. Measurement of Single-Molecule Resistance by Repeated Formation of Molecular Junctions. *Science* **301**, 1221–1223 (2003).
3. Li, Y., Xiang, L., Palma, J. L., Asai, Y. & Tao, N. Thermoelectric effect and its dependence on molecular length and sequence in single DNA molecules. *Nat. Commun.* **7**, 11294 (2016).

Comment 4

The words in Figure 3 were not clear.

Reply 4

In the manuscript, we modified the caption as follow to increase clarity:

“Figure 3. Histogram of the theoretical phononic contribution κ_{ph} to the total thermal conductance of the junction (top) vs. experimental total thermal conductance κ_j (bottom). In the top panel, the results obtained from MD simulations for the phononic thermal conductance of the DHBT-OPE3-An molecule are represented in the form of a histogram. The structure of the DHBT-OPE3-An molecule is also shown (carbon atoms in grey, hydrogen atoms in white, sulfur atoms in

yellow). In the bottom panel, the black solid line is the weighted average between the two experiments. The green shaded area represents the uncertainty on the weighted average. For comparison with the experimental values, the cyan dashed line represents the total predicted thermal conductance κ_{TOT} obtained as a sum of the theoretical phonon conductance κ_{ph} (i.e. the average of the theoretical histogram on top) plus the theoretical electronic contribution κ_{el} predicted by means of density functional theory.”

Reviewer: 2

Comment 1

There are typos should be carefully checked and revised, for example, the subtitle should be “Introduction”; the line 173, what does the ‘F’ represent for, and so on.

Reply 1

We thank the reviewer for reading the manuscript despite the formatting errors, which were fixed in the revision. A native speaker has carefully read the manuscript.

Comment 2

Why the author choses the oligo(phenyleneethynylene)-9,10-anthracenyl single molecule junction with dihydrobenzo[b]thiophene anchoring groups to do the calculation and get experimentally the full zT. Can this method be applied to another general molecules?

Reply 2

We thank again the reviewer for their comment, and we are grateful for the opportunity to better clarify what we believe has been a concern also for the other reviewers. The reasons to select the DHBT-OPE3-An are manifold. The main reason is practical and relates to the ability of the DHBT-OPE3-An to form single molecule junctions with relative high yield and to be able to sustain high temperature differences between the two electrodes (cf. *Dekkiche et al. Nanoscale* **12** (36): 18908–17, (2020)). Given the complexity of the experiment this is already a strong point. Another reason is due to the peculiar chemical structure of the molecule itself. In fact, by selecting a derivative of the OPE3 family, we have the opportunity to study the role of anchoring groups (DHBT) and side group (An) for phonon engineering, while keeping open the possibility to benchmark such a molecule against a large variety of experimental data already present in the literature. This is done in a separate Supplementary Information file, that we attach together with the revised version of the manuscript. There, Figure S3 shows the comparison of the experimental results obtained in this work with other data in the literature.

These reasons were also summarized in the manuscript:

*“Given the very tedious and slow thermal transport measurements, this particular molecule was chosen as to address some of the most important current questions in the field. Namely, the interesting aspect of this particular molecule is the increase of vibrational states by locally adding atoms to only one (the central) part of the molecule with respect to the reference OPE3 molecule. This is an important step towards the inclusion of moieties to enhance or reduce the thermal conductance of a molecule through localization and quantum interference effects [Gotsmann et al. *Applied Physics Letters* 120 (16): 160503, (2022)]. Furthermore, the attachment of side-groups in the central part of a molecule is a starting point of cross-linking strategy of molecular layers [Dekkiche et al., *Nanoscale* 12 (36): 18908–17, (2020)]. In addition, we study, in the thermal context, the*

enhanced binding yield reported for DHBT end groups, which may reduce the systematic risk of statistical approaches. These data then motivate further studies with series of molecules systematically varying only one aspect, which is commonly done for the simpler electrical conductance and thermoelectric measurements.”

As we also write in the paper, we do not foresee any issue in applying the same protocol to other molecules belonging to the OPE3 family or more diverse structures. As already shown in *Mosso et al. [Nano Letters 19 (11): 7614–22, (2019)]*, within the same protocol the thermal conductance of alkane chains (octane-dithiol) can be measured, and as confirmed in *Dekkiche et al. [Nanoscale 12 (36): 18908–17, (2020)]* also the thermoelectric properties of different OPE3 derivatives can be tested.

Comment 3

The author should clarify the novelty of this manuscript and what developments of science is this manuscript contributed to.

Reply 3

We thank the reviewer for their comment.

The main novelties pointed out in the manuscript can be summarized in the following points, as given in the revised manuscript:

1. *“Our results represent, to our knowledge, the first ever experimental measurement of the complete zT of a single molecule at room temperature and within the same setup, including the phononic contribution to the total thermal conductance.”* The importance of the combination of these measurements is explained in the revised introduction (see above)
2. *“This work also confirms that phonon engineering is possible in single molecules. Indeed, it is interesting to notice that despite an increase of 30% in the number of atoms and degrees of freedom, the thermal conductance of the DHBT-OPE3-An only increases by about 10% compared to the predicted 22 pW/K for the DHBT-OPE3-Ph analog, hinting at the influence of side groups.”* This is an important conjecture [cf. *Gotsmann et al. Applied Physics Letters 120 (16): 160503, (2022)*].
3. The new protocol allows for the study of the role of anchoring and side groups in vision of possible applications. *“Within the accuracy of the experimental and theoretical results presented here, the DHBT-OPE3-An possesses a thermal conductance that is slightly lower than average despite the improved cleanliness of the binding.”* This is particularly interesting in vision of thermoelectric applications.

Comment 4

The author claimed the experimental result is in excellent agreement with density functional theory and molecular dynamics predictions. More experimental details should be given in the manuscript, otherwise, it is insufficient to judge whether the method is reasonable or not.

Reply 4

We thank the reviewer for the comment, and we agree we should provide more details about the experiment as well as about the theoretical predictions. We added a significant amount of detail to the revised manuscript, including typical operational parameters and comparison to protocols in the literature. For the sake of readability, we added some of the extra information in a separate Supplementary Information file (SI), which we attach with the improved version of the manuscript. In the SI file, we give more experimental details on the method used to extract the thermal

conductance κ , as difference of linear fits. Also, we provide the phononic heat flux spectrum obtained by MD simulations and we comment more on the rigidity of the molecular backbone.

Reviewer: 3

Comment 1

The authors need to clarify the concept of the simultaneous measurement in thermoelectric parameters with a detailed explanation compared to Phys. Rev. B 2011, 83, 113305.

Reply 1

We thank the reviewer for their comment. Our method for the simultaneous measurement of the thermoelectric parameters is based on the one described by *Widawsky et al.* in *Nano Letters*, 12 (1): 354–58, 2012.

Compared to Phys. Rev. B 2011, 83, 113305 we instead highlight the following differences:

1. Our method is able to measure samples with the size of \sim nm, in case of single molecules, or even below in case of single atoms. Other methods in literature hardly reach those limits.
2. Our suspended MEMS structure is provided with long and thin beams, which guarantee high thermal insulation ($R_{\text{therm}} \sim 10^7 - 10^8$ K/W) that doesn't make necessary the use of four-probe measurement for the thermal signal.
3. The design of our sensor/heater allows for the simultaneous characterization of the electrical and thermal properties even for electrically conductive samples.

These points are summarized in the revised manuscript:

"We note, the junction thermal conductance κ_j is so small that it dominates the entire thermal path even though the already low values of κ_{MEMS} . Therefore, thermal four-probe techniques as described for example in Okada et al. [Phys. Rev. B 2011, 83, 113305, (2011)] can be avoided. On a molecular scale no convincing approach for a four-probe measurement (neither electrically nor thermally) has been proposed. One therefore talks about the junction conductance rather than the conductance of a molecule."

Other break junction methods present in the literature, like the mechanically controllable break junction (MCBJ), offer the possibility to measure the electronic properties of a single molecule, but only to indirectly estimate the thermal conductance (and in particular the phononic contribution) of single molecule junctions [cf. *Gehring et al. Nature Nanotechnology* 16 (4): 426–30, 2021].

Furthermore, we want to point out that only one other group in the world has been able to measure, so far, the thermal conductance of single molecules. Within their method, they use a complementary approach to ours, where the sensor and heater are placed on the scanning tip, rather than on the counter electrode [cf. *Cui et al. Nature* 572 (7771): 628–33, 2019].

Comment 2

To confirm the system represented in this research, they show the data with another single thermoelectric molecule.

Reply 2

We thank the reviewer for their comment. As already stated in the Reply 2 for Reviewer 2, the method has already been tested against different molecules, independently for the measurement of the thermal conductance [cf. Mosso et al. Nano Lett. 19, 7614–7622 (2019)] and for the measurement of the Seebeck coefficient [cf. Dekkiche et al., Nanoscale 12 (36): 18908–17, (2020)]

Only the duration and complexity of the experiment did not allow us to include another molecule in the study, but we do not see other obstacles in applying the full method to different samples.

We improved the manuscript by explaining this in the section starting with: *“Given the very tedious and slow thermal transport measurements, this particular molecule was chosen as to address ...”*

Comment 3

In terms of the lattice contribution of the thermal conductivity, they need to represent the experimental data compared with MD simulation.

Reply 3

We agree that a careful and clear comparison of simulation and experimental data is necessary. This is done in figure 3 of the revised main manuscript, the capture of which reads:

“Histogram of the theoretical phononic contribution κ_{ph} to the total thermal conductance of the junction (top) vs. experimental total thermal conductance κ_J (bottom). In the top panel, the results obtained from MD simulations for the phononic thermal conductance of the DHBT-OPE3-An molecule are represented in the form of a histogram. The structure of the DHBT-OPE3-An molecule is also shown (carbon atoms in grey, hydrogen atoms in white, sulfur atoms in yellow). In the bottom panel, the black solid line is the weighted average (avg) between the two experiments. The green shaded area represents the uncertainty (unc) on the weighted average. For comparison with the experimental values, the cyan dashed line represents the total predicted thermal conductance κ_{TOT} obtained as a sum of the theoretical phonon conductance κ_{ph} (i.e., the average of the theoretical histogram on top) plus the theoretical electronic contribution κ_{el} predicted by means of density functional theory.”

Furthermore, we added theoretical results in the Supplementary Information file, that we attach together with the following rebuttal letter and the revised manuscript. The SI contains also a comparison of the experimental and theoretical data of other molecules of the OPE3 family.

REVIEWER COMMENTS

Reviewer #1 (Remarks to the Author):

The manuscript has been revised and can be accepted.

Reviewer #2 (Remarks to the Author):

The manuscript has been revised carefully and can be accepted now.

Reviewer #3 (Remarks to the Author):

The authors have worked on the revision and coped well with most comments. However, the authors should at least verify this protocol with the molecules in Nature 572, 628–633 (2019), Nano Lett. 19, 7614–7622 (2019) that they presented in the introduction section. Thus, I'm afraid I have to disagree this manuscript meets the requirement for publication of NCMM.

Reviewer 3 comment:

The authors have worked on the revision and coped well with most comments. However, the authors should at least verify this protocol with the molecules in Nature 572, 628–633 (2019), Nano Lett. 19, 7614–7622 (2019) that they presented in the introduction section. Thus, I'm afraid I have to disagree this manuscript meets the requirement for publication of NCMM.

Reply

We are resubmitting this manuscript now containing data and discussion on the verification using the SAc-OPE3 molecule, which makes the desired link to the work previously published in 2019 on thermal transport. (The other molecule possible for comparison of protocol would be the octanedithiol. However, while the agreement with thermal conductance measurement is reached, there is not an established agreement on the electrical conductance of these molecules. Therefore, the SAc-OPE3 appears a better choice, because there is plenty of literature on Seebeck and electrical conductance which is widely agreed upon.)

The experiments were successful, and we added these data in the manuscript.

We added to the manuscript in the abstract:

“The protocol is verified using SAc-OPE3, for which individual measurements for its transport properties exist in the literature.”

In the introduction:

“In addition, the protocol is verified using SAc-OPE3, for which individual measurements for its transport properties exist in the literature.”

In the experimental section:

“This protocol has been applied also to OPE3 with SAc end groups, a molecule that has been studied using the same setup in thermal conductance measurements (see supplementary information).”

In the discussion section:

“The protocol and setup described here has been verified carefully in its individual measurements. In particular, the thermal and electrical conductance measurement technique was previously independently verified by two groups [Mosso 2019, Cui 2019], and the Seebeck coefficient measurement was verified using SAc-OPE3 in comparison to literature (see supplementary information). The SAc-OPE3 molecule yields $zT \approx 2 \cdot 10^{-5}$.”

And in the Supplementary Information:

“Reference measurement using SAc-OPE3

In order to verify the protocol to measure the Seebeck coefficient, it was also applied to the same molecule that was previously examined in the same setup, namely the OPE3 molecule with standard thioacetate end groups (SAc-OPE3), for which the thermal conductivity was reported by Mosso et al.⁷ in agreement with observations by Cui et al.⁸. The protocol to measure Seebeck coefficient was applied also to this molecule yielding $7.9 \pm 4 \mu\text{V}/\text{K}$. This is in agreement with literature values obtained for single molecules and self-assembled monolayers, namely $8.0 \pm 0.8 \mu\text{V}/\text{K}$ ¹⁷, $7.78 \pm 0.34 \mu\text{V}/\text{K}$ ¹⁸, and $9.7 \pm 0.2 \mu\text{V}/\text{K}$ ¹⁹. This makes SAc-OPE3 the second molecule for which the entire zT was measured using this protocol, yielding $zT \approx 2 \times 10^{-5}$.”

REVIEWERS' COMMENTS

Reviewer #1 (Remarks to the Author):

The manuscript has been revised, and I think it can be accepted.